**Subject Category:**
Biology (whole organism)

health and disease and epidemiology/
computational biology

human ectoparasite transmission, *Yersinia pestis*,
serial interval, reproduction number,
Third Pandemic

**Author for correspondence:**
Katharine R. Dean
e-mail: k.r.dean@ibv.uio.no

# Epidemiology of a bubonic plague outbreak in Glasgow, Scotland in 1900

Katharine R. Dean, Fabienne Krauer and Boris V. Schmid

Department of Biosciences, Centre for Ecological and Evolutionary Synthesis (CEES), University of Oslo, 0316 Oslo, Norway

KRD, 0000-0003-2262-0385

On 3 August 1900, bubonic plague (*Yersinia pestis*) broke out in Glasgow for the first time during the Third Pandemic. The local sanitary authorities rigorously tracked the spread of the disease and they found that nearly all of the 35 cases could be linked by contact with a previous case. Despite trapping hundreds of rats in the area, there was no evidence of a rat epizootic and the investigators speculated that the outbreak could be due to human-to-human transmission of bubonic plague. Here we use a likelihood-based method to reconstruct transmission trees for the outbreak. From the description of the outbreak and the reconstructed trees, we infer several epidemiological parameters. We found that the estimated mean serial interval was 7.4–9.2 days and the mean effective reproduction number dropped below 1 after implementation of control measures. We also found a high rate of secondary transmissions within households and observations of transmissions from individuals who were not terminally septicaemic. Our results provide important insights into the epidemiology of a bubonic plague outbreak during the Third Pandemic in Europe.

## 1. Introduction

Plague is a zoonotic disease, caused by the bacterium *Yersinia pestis*, which is well known as the cause of at least three pandemics in human history: the First Pandemic (sixth to eighth centuries), the Second Pandemic (fourteenth to nineteenth centuries) and the Third Pandemic (beginning in the nineteenth century). At the beginning of the Third Pandemic, *Y. pestis* spread from Asia to Europe, Africa, Australia and the Americas along maritime transport networks [1]. These introductions led to the establishment of plague reservoirs in rodent populations around the world, which today pose a recurrent threat to nearby human populations [2].

The most common form of plague infection in humans is bubonic plague, caused by the bite of an infected flea vector [3,4]. Today, cases of bubonic plague typically arise through contact with sylvatic or commensal animals and their fleas [3,4]. In the

past, large epidemics of plague in Asia were caused by epizootics in the susceptible urban rat population, which led infected rat-flea vectors to seek alternative mammalian hosts [5]. However, there is some evidence that bubonic plague may also spread between people through human ectoparasite vectors such as body lice (*Pediculus humanus humanus*) or human fleas (*Pulex irritans*). This is supported by experimental and epidemiological studies that have shown that human ectoparasites are potential vectors for plague and have been found infected during modern outbreaks in Africa [6–9].

In general, the epidemiology of plague outbreaks in Europe is poorly understood [10]. Even though there were hundreds of plague notifications during the Third Pandemic, research on the disease in Europe has mainly focused on the large outbreaks during the Second Pandemic. However, records from mediaeval and early-modern Europe provide limited information about the nature of the outbreaks and lack the scientific awareness of the bacterium and its transmission that was formed during the investigation of plague outbreaks in India at the end of the nineteenth century. Therefore, there is an opportunity to better understand the epidemiology of plague outbreaks in Europe during the Third Pandemic. Although these outbreaks cannot simply be assumed to be representative of the Second Pandemic, they can provide a valuable point of comparison for future studies.

Here we use an official government report of plague in Glasgow, Scotland in 1900 to study the epidemiology of plague in Europe [11]. During this remarkably well-documented outbreak, investigators observed that many cases of plague could be linked by contact with a previous case and they found no evidence of a rat epizootic. The information in the report can be used to partially reconstruct the transmission tree; however, some transmission events are not known. To address this problem, we applied a robust likelihood-based method to reconstruct probable transmission trees, from which we estimated several disease transmission parameters [12].

For disease spread at an individual level, we estimated the serial interval, which is defined as the time between the symptom onset of a case and the symptom onset of their infector [13]. To understand how the disease spreads on a scale of disease generations, we calculated the effective reproduction number $R_e$ defined as the average number of secondary cases produced by a primary case [13]. We compared $R_e$ before and after notification of the disease to assess the impact of intervention measures on controlling the outbreak. Finally, we discuss different aspects of transmission, including the number of secondary cases arising within the same household and the possibility of those arising from individuals who ultimately recovered from the disease (non-septicaemic transmission).

# 2. Material and methods

## 2.1. Description of the outbreak

On 25 August 1900, the sanitary authorities of Glasgow were notified of several suspected cases of bubonic plague, despite no known cases of plague in Britain at the time [14]. By the following day, they confirmed their initial diagnosis of *Y. pestis* infection from cultures taken on glycerin agar, and later in the week by animal experiments at the University of Glasgow [14]. Upon the identification of the plague, the Medical Officer of Health in Glasgow opened an immediate investigation into the spread of the disease. The investigation led to the identification of the index cases, known as Mrs B., a fish hawker, who sickened along with her granddaughter, on 3 August (Day 0 of the outbreak) [14]. The sanitary authorities searched for contacts associated with Mrs B. or who had attended her wake, leading to the examination and quarantine of more than one hundred people in a 'reception house' for observation [14].

In addition to contact tracing and quarantining, the sanitary authorities implemented several other measures to control the spread of plague including (1) removal of cases to the hospital, (2) cessation of wakes for deaths attributed to plague, (3) fumigation of infected homes with liquid sulfur dioxide and disinfection with a formalin solution, (4) removal and treatment of clothing and sheets, (5) disinfection of all homes and communal areas in infected tenements with chloride of lime (chlorine powder) solution, (6) emptying of ashpits and (7) dissemination of information about the disease to the public and health professionals [15].

Two years prior to the outbreak in Glasgow, Paul-Louis Simond had discovered that rats and their fleas could transmit plague to humans [16]. Consequently, the sanitary authorities in Glasgow were particularly interested in the role of rats in spreading the disease. They noted that rats were numerous in the infected tenements; however, there was no evidence that the mortality among rats was abnormal [15]. The authorities undertook an extensive trapping and extermination campaign, which included the examination of 326 rats [11,17]. Despite their efforts, they found no evidence of plague in

the rat population at any time during the outbreak, leading them to conclude that plague may have spread directly between humans through clothing among other means, and possibly by 'the suctorial parasites of mankind' [11]. Notably, rats were caught and examined for plague in Glasgow during the period between 1900 and 1907, and a small number of infected rats were found in the years after the 1900 outbreak: 1901 (122 of 1641), 1902 (30 of 6492) and 1907 (1 of 140) [17].

In the official report of the outbreak published in 1901, the local authorities identified 37 cases of plague in and around Glasgow between 3 August and 24 September 1900 [11]. By March 1901, the city had a population of 761712, but the cases were primarily located in the densely populated Gorbals area, on the south bank of the river Clyde [11]. Most of the cases after notification were identified as a primary bubonic or septicaemic plague by the presence of external buboes [11]. However, we excluded one of these confirmed cases, called 'Govan boy', for whom there was no case information [11]. The additional suspected case presented with primary pneumonia, but it was noted that the survival of the patient and failure to retrieve the bacteria discredited the assumption of plague pneumonia [11]. Thus, our analysis included 35 cases with information about their date of symptom onset and possibly their contacts with previous cases. We broadly defined a contact to be any individual that lived at the same address as the case; any individual who visited the house of a case; or any individual who provided formal or informal care to the case.

## 2.2. Likelihood of possible transmission pairs and estimation of the serial interval distribution

Using the notation in Hens *et al.* [12], we assigned each case a unique case identifier (*i*) [12]. We numbered the cases by the symptom onset date ($t_i$) and if the symptom onset dates were equal we used the original order from the case reports [12]. For each case *i*, except the index cases, we denoted the unique infector as $v(i)$ or contacts as $w(i)$, if known. With no missing information for $v(i)$, the serial interval can be calculated as a positive number for each case *i* as $t_i - t_{v(i)}$, which is the difference between the symptom onset of case *i* and the symptom onset of the infector $v(i)$. The observed serial intervals can be used to describe the serial interval distribution $g(t_i - t_{v(i)}|\theta)$ and the effective reproduction number, $R_e$. However, for the outbreak in Glasgow, the transmission tree is not fully resolved, and information about the infectors is often missing.

To find the missing transmission pairs, we used the method in Hens *et al.* [12], which finds the probability $p_{ij}(v, w)$ that case *i* was infected by case *j*, given the estimated serial interval distribution (described below), and given any prior information on the infectors in $v$ ($1 \times n$ matrix) and the contacts $w$ ($n \times n$ matrix). The total log-likelihood of the data is then given by summing the total log-likelihood of all cases, excluding the index cases,

$$E\{\ell(\theta|t, v, w)\} = \sum_{i=3}^{n} \sum_{j=1}^{n} p_{ij}(v, w) \log g(t_i - t_j|\theta). \tag{2.1}$$

We assumed a gamma distribution to describe the probability density of the serial interval distribution for bubonic plague. Maximizing the expected log-likelihood yields estimates for the parameter set $\hat{\theta} = \{a, b\}$, where *a* is the shape parameter and *b* is the scale parameter of the gamma distribution.

The probability that case *i* was infected by case *j*, $p_{ij}$, is the product of the probability of observing the serial interval between two cases, $g(t_i - t_j|\theta)$, and the probability of an infectious contact between *i* and *j*, $\pi_{ij}(v, w)$, normalized by the probability of case *i* being infected by any other case *k*,

$$p_{ij}(v, w, \hat{\theta}) = \frac{g(t_i - t_j|\hat{\theta}) \times \pi_{ij}(v, w)}{\sum_{k \neq i} g(t_i - t_k|\hat{\theta}) \times \pi_{ik}(v, w)}. \tag{2.2}$$

The probability of an infectious contact between cases *i* and *j*, $\pi_{ij}(v, w)$, is informed by the contact information collected during the outbreak, such that:

— $\pi_{ij}(v, w) = 1$, if case *j* is the only possible infector of case *i*;
— $\pi_{ij}(v, w) = 1/m$, if case *j* is one of *m* contacts and a possible infector of case *i*;
— $\pi_{ij}(v, w) = 1/(i - 1)$, if there are no contacts for case *i* and it is not an index case.

We used the prior-based expectation maximization (PEM) algorithm described by Hens *et al.* [12] to obtain the maximum expected log-likelihood value [12]. By this process, the probability of infectious contacts based on information collected during the outbreak is evaluated first (P-step), then the probability of transmission is evaluated given the current estimate of the serial interval parameters $\theta$

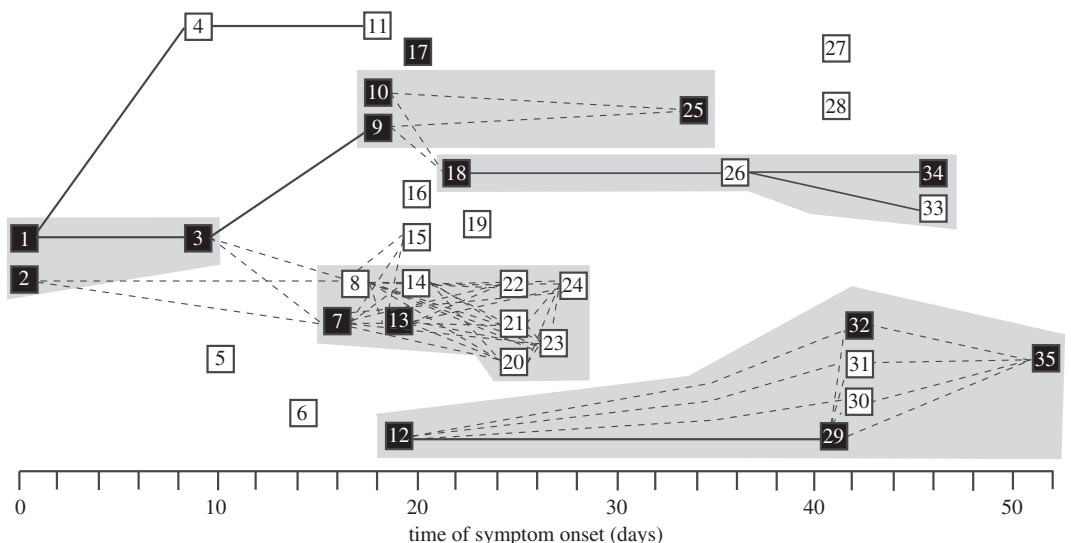

**Figure 1.** Recorded transmission events during a plague outbreak in Glasgow, Scotland, from 3 August 1900 to 24 September 1900. Cases are represented by squares (solid = dead) and ordered by the date of symptom onset. Solid lines indicate transmission events between cases with a known infector. For cases without a known infector, dashed lines indicate reported contacts between cases. Grey shaded boxes indicate cases in the same household.

(E-step), and then the parameters of $\theta$ are found that maximize the likelihood given the probabilities of transmission (M-step), repeating the E-step and M-step until the results converge to the maximum log-likelihood estimate [12].

To examine the effect of potentially false information for the known pairs on the estimated serial interval distribution, we repeated the analysis by leaving out information for the infector $v(i)$ for each pair one by one. The resulting change in the expected log-likelihood estimate for the parameter set $\theta$ for case $i$ is called the 'global influence measure' and can be written as $GI_i = E\{\ell(\hat{\theta}_{[-i]})\} - E\{\ell(\hat{\theta})\}$ [12]. Additionally, we considered the extreme case that all recorded contact information was unreliable and repeated the PEM algorithm using only the symptom onsets. We also considered the scenario that only moribund cases, with high levels of septicaemia, were capable of infecting vectors and we repeated the analysis restricting the possible infectors to those that died from the plague.

## 2.3. Reconstruction and analysis of possible transmission trees

From the likelihood procedure, we obtained probabilities that any case $i$ was infected by any case $j$. Using these probabilities to specify a multinomial distribution, we sampled a single infector $v(i)$ for each case $i$ (excluding the index cases) to produce a fully reconstructed transmission tree. We repeated this process to produce 1000 possible transmission trees for each model. For each simulated tree, we calculated the average serial interval for all cases, for household transmission, and for community transmission. For the trees simulated from the model that allowed for any individual to be an infector, we calculated the number of secondary cases produced by each case. We calculated the effective reproduction number as the average secondary infections for cases with symptom onsets on day $t$: $R_e(t) = \sum_j \sum_{i=} p_{ij}(v, w, \hat{\theta})$. Additionally, we counted the number of cases with infectors in the same household and the number of cases with infectors that ultimately survived their infections (i.e. that spread the disease without being terminally septicaemic).

## 3. Results

Thirty-one (88%) suspected cases of plague in Glasgow were diagnosed by the presence of external buboes; and 17 (48.5%) of these cases were confirmed by bacteriological examination [11]. The median patient age was 20 years (range less than 1–60 years): 21 (60%) of the cases were female and 14 (40%) were male. The case-fatality rate for the outbreak was 42.8% for both men and women. From the 15 fatal cases, we found that the median symptomatic period was 6 days (range 2–44 days). There was not enough information in the patient histories to calculate the symptomatic periods for non-fatal cases.

The observed transmission tree for the outbreak is shown in figure 1. The report included contact information for 24 (69%) of the cases; and for 8 of these, they identified a single known infector. From

the eight observed pairs, we found that the mean serial interval was 11.5 days (95% confidence interval (CI): 9.0, 20.6) (figure 2a).

Using the likelihood-based method, we obtained the probabilities (table 1) for the missing transmission pairs based on the date of symptom onset and the contact information. To check the influence of the known serial intervals on the results, we calculated the global influence measure for the observed pairs, shown in table 2. We found that one pair (case 29-case 12) had a relatively high GI measure, but the impact of this pair on the mean serial interval was negligible.

To estimate the serial interval for the outbreak, we used the probabilities from the likelihood-based approach to simulate transmission trees for different models. The mean serial intervals estimated from the simulated trees were 7.4 days (95% CI: 6.5, 8.6) assuming non-terminal cases could transmit and 9.2 days (95% CI: 7.9, 10.6) assuming only terminal cases could be infectors (figure 2b and figure 3). There were no significant differences between the average serial intervals for household and community transmissions across the models (figure 3).

From the simulated trees allowing non-terminally ill infectors, we estimated the time course reproduction numbers. We found that the effective reproduction number declined throughout the duration of the outbreak, shown in figure 2c. Before notification of the outbreak on day 22, the average reproduction number was 1.6 (95% CI: 0.9, 2.9). Following notification and implementation of control measures, the average reproduction number was 0.6 (95% CI: 0.0, 2.5).

We also estimated the proportion of secondary household transmissions and the proportion of transmissions from non-septicaemic infections (figure 4). From the observed data, we found from that 62.5% of infections occurred between household contacts. Using both the symptom onset dates and the contact information, we found that the proportion of secondary household infections was 51.5% (95% CI: 51.5, 51.5). When simulating trees using only the symptom onset data and ignoring known contact information, we estimated that 24.4% (95% CI: 18.1, 34.6) of the transmission pairs occurred within a household (figure 4a). Next, we identified transmission pairs where the infector had a non-lethal infection. Based on the eight known pairs in the data, 37.5% of cases were infected by persons who survived their infection (non-septicaemic transmission). The proportions of non-septicaemic transmission were 51.7% (95% CI: 39.3, 66.6) and 38.9% (95% CI: 27.3, 48.6), using the trees with and without contact information, respectively (figure 4b).

## 4. Discussion

Our study reports on the epidemiological characteristics of an outbreak of bubonic plague in Glasgow in 1900. From the information in the report, we found that the symptomatic period for bubonic plague in fatal cases was 6 days, which agrees well with the estimate of 5.5 days reported for 100 fatal cases in India [18]. The case-fatality rate was around 40% and this is consistent with other reports of bubonic plague in the pre-antibiotic era [4]. These estimates support the diagnosis of bubonic plague made by the sanitary officials.

We used the contact-tracing information from the official report and applied a likelihood-based method to infer plausible transmission trees. With the reconstructed trees, we directly inferred the serial interval and the effective reproduction number for the outbreak. We estimated that the mean serial interval was on average 7.4–9.2 days (95% CI: 6.5, 10.6), depending on the model assumptions, which was shorter than the mean observed serial interval of 11.5 days (95% CI: 9.0, 20.6). The difference in the means, although not significant, could be attributed to the small number of observed serial intervals or a bias towards observing longer intervals. To our knowledge, there are no other estimates of serial intervals for bubonic plague, thus the reliability of either estimate is difficult to assess. The serial interval for a vector-borne disease is longer than for directly transmitted diseases because they include time in the host as well as in the vector. Given that bubonic plague is transmitted by vectors and that Y. pestis can be cultivated from the serum on average 5 days post-infection, and as early as 2 days, an estimate of one to two weeks seems biologically plausible [19].

The reproduction number decreased after notification of the disease. Our estimate of 1.6 before notification is within the range reported (1.4–1.8) for nine outbreaks of plague in Europe during the Second Pandemic with suspected human ectoparasite transmission [20]. The small size and short duration of the outbreak suggest that quarantining and sanitation were effective in stopping the spread of plague, which is also reflected in the drop in $R_e$ below 1 after the implementation of control measures.

Many studies have reported household clustering of cases during Second Pandemic plague outbreaks in Europe [21–26]. For Glasgow, we found that more than half of the secondary cases arose from

**Table 1.** The most likely infectors and their probability according to the likelihood procedure based on the time of symptom onset (EM algorithm), the time of symptom onset augmented with the contact information (PEM algorithm) and the time of symptom onset augmented with the contact information and with only terminally ill infectors (PEM algorithm). Source cases for which the probability was lower than 0.1 were omitted from the table.

| case ($i$) | likely infectors $j$ based on symptom onset | likely infectors $j$ based on symptom onset and contacts | likely infectors $j$ based on symptom onset and contacts (only terminal infectors) |
|---|---|---|---|
| 5 | v1, v2 (0.274) | v1, v2 (0.192) | v1, v2 (0.285) |
|   | v3, v4 (0.225) | v3, v4 (0.307) | v3 (0.428) |
| 6 | v3, v4 (0.273) | v3, v4 (0.277) | v1, v2 (0.218) |
|   | v5 (0.276) | v5 (0.299) | v3 (0.563) |
| 7 | v3, v4 (0.220) | v2 (0.185) | v1, v2 (0.207) |
|   | v5 (0.240) | v3 (0.818) | v3 (0.584) |
|   | v6 (0.203) | | |
| 8 | v3, v4 (0.183) | v2 (0.178) | v1, v2 (0.204) |
|   | v5 (0.203) | v3 (0.821) | v3 (0.591) |
|   | v6 (0.224) | | |
|   | v7 (0.116) | | |
| 10 | v3, v4 (0.144) | v3, v4 (0.117) | v1, v2 (0.104) |
|    | v5 (0.163) | v5 (0.136) | v3 (0.310) |
|    | v6 (0.211) | v6 (0.210) | v7 (0.480) |
|    | v7 (0.167) | v7 (0.206) | |
|    | v8 (0.103) | v8 (0.161) | |
| 12 | v5 (0.107) | v6 (0.133) | v1, v2 (0.104) |
|    | v6 (0.155) | v7 (0.148) | v3 (0.310) |
|    | v7 (0.149) | v8 (0.141) | v7 (0.480) |
|    | v8 (0.124) | v9, v10, v11 (0.110) | |
| 13 | v5 (0.107) | v7 (0.512) | v7 (1.0) |
|    | v6 (0.155) | v8 (0.487) | |
|    | v7 (0.149) | | |
|    | v8 (0.124) | | |
| 14 | v6 (0.115) | v7 (0.492) | v7 (0.532) |
|    | v7 (0.123) | v8 (0.507) | v13 (0.467) |
|    | v8 (0.117) | | |
| 15 | v6 (0.115) | v7 (0.492) | v7 (0.532) |
|    | v7 (0.123) | v8 (0.507) | v13 (0.467) |
|    | v8 (0.117) | | |
| 16 | v6 (0.115) | v7 (0.113) | v7 (0.177) |
|    | v7 (0.123) | v8 (0.117) | v9 (0.179) |
|    | v8 (0.117) | v9, v10, v11 (0.111) | v12, v13 (0.155) |
| 17 | v6 (0.115) | v7 (0.113) | v7 (0.177) |
|    | v7 (0.123) | v8 (0.117) | v9 (0.179) |
|    | v8 (0.117) | v9, v10, v11 (0.111) | v12, v13 (0.155) |
| 18 | ($<$0.100) | v9, v10 (0.500) | v9, v10 (0.500) |

(*Continued.*)

| case (*i*) | likely infectors *j* based on symptom onset | likely infectors *j* based on symptom onset and contacts | likely infectors *j* based on symptom onset and contacts (only terminal infectors) |
|---|---|---|---|
| 19 | ($<$0.100) | ($<$0.100) | v7 (0.111) |
| | | | v9, v10 (0.131) |
| | | | v12, v13 (0.139) |
| | | | v17 (0.143) |
| | | | v19 (0.122) |
| 20 | ($<$0.100) | v7(0.188) | v7 (0.428) |
| | | v8 (0.217) | v13 (0.571) |
| | | v13 (0.281) | |
| | | v14 (0.312) | |
| 21 | ($<$0.100) | v7(0.188) | v7 (0.428) |
| | | v8 (0.217) | v13 (0.571) |
| | | v13 (0.281) | |
| | | v14 (0.312) | |
| 22 | ($<$0.100) | v7(0.188) | v7 (0.428) |
| | | v8 (0.217) | v13 (0.571) |
| | | v13 (0.281) | |
| | | v14 (0.312) | |
| 23 | ($<$0.100) | v13 (0.124) | v7 (0.419) |
| | | v14 (0.142) | v13 (0.580) |
| | | v20, v21, v22 (0.187) | |
| 24 | ($<$0.100) | v14 (0.111) | v7 (0.415) |
| | | v20, v21, v22 (0.177) | v13 (0.584) |
| | | v23 (0.131) | |
| 25 | v23 (0.113) | v9, v10 (0.500) | v9, v10 (0.500) |
| | v24 (0.126) | | |
| 27 | v25 (0.212) | v25 (0.230) | v18 (0.105) |
| | v26 (0.224) | v26 (0.255) | v25 (0.476) |
| 28 | v25 (0.212) | v25 (0.230) | v18 (0.105) |
| | v26 (0.224) | v26 (0.255) | v25 (0.476) |
| 30 | v25 (0.161) | v29 (0.945) | v12 (0.113) |
| | v26 (0.175) | | v29 (0.886) |
| 31 | v25 (0.161) | v29 (0.945) | v12 (0.113) |
| | v26 (0.175) | | v29 (0.886) |
| 32 | v25 (0.161) | v29 (0.945) | v12 (0.113) |
| | v26 (0.175) | | v29 (0.886) |
| 35 | v30, v31, v32 (0.101) | v29 (0.220) | v29 (0.471) |
| | v33, v34 (0.158) | v30, v31, v32 (0.259) | v32 (0.528) |
| mean | 8.28 | 7.4 | 9.2 |
| (95% CI) | (6.81, 9.72) | (6.48, 8.63) | (7.9, 10.6) |

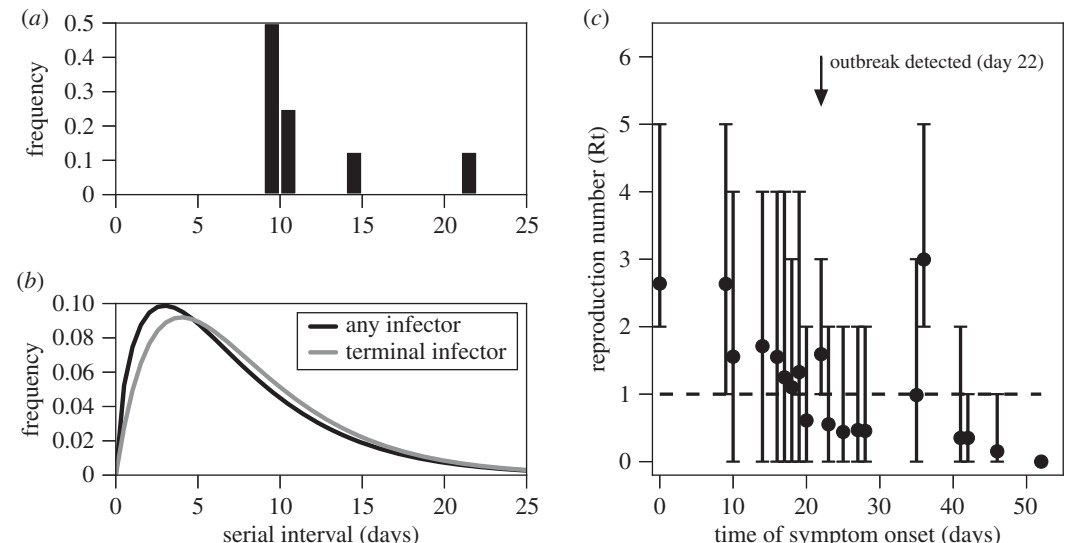

**Figure 2.** Reconstruction of transmission events for a plague outbreak in Glasgow, Scotland, from 3 August 1900 to 24 September 1900. (*a*) Relative frequency of the serial intervals, based on eight observed transmission events, (*b*) Relative frequency of the serial intervals, based on 8 observed transmission events and 27 reconstructed transmission events. The black line shows the distribution with any infector, mean = 7.4 days [95% CI: 6.5, 8.6]. The grey line shows the distribution with only terminally ill infectors, mean = 9.2 days [95% CI: 7.9, 10.6]. (*c*) Average effective reproduction number ($R_e(t)$) per day (dots) and 95% bootstrap percentile confidence interval (bars).

**Table 2.** Global influence of the observed serial intervals.

| case (*i*) | infector (*v(i)*) | global influence measure (*GIᵢ*) |
|---|---|---|
| 3 | 1 | 0.0 |
| 4 | 1 | 0.0 |
| 9 | 3 | 0.42 |
| 11 | 4 | 0.42 |
| 26 | 18 | 0.46 |
| 29 | 12 | 2.85 |
| 33 | 26 | 0.49 |
| 34 | 26 | 0.49 |

infectors in the same household, which was higher than expected based only on the symptom onsets (figure 3*a*). Household clustering of plague cases in historical outbreaks may be attributed to pneumonic plague, which spreads directly between people [23]. However, our results show that a high rate of secondary transmission within households can also occur during bubonic outbreaks. A similar finding was reported for a plague outbreak in Nepal in 1967, with suspected human ectoparasite transmission [27]. By contrast, household clustering was not a feature of plague epidemics spread by rats, as observed in Bombay, Sydney and New Orleans [28–30].

For many vector-borne diseases, like plague spread by rats, it may be difficult or impossible to trace successive cases and establish transmission chains. However, human ectoparasites are tightly associated with their hosts or host environment, and switching hosts may require close and prolonged contact, such as staying in the home or sharing clothes [31,32]. Under these conditions, the transmission of bubonic plague through a human ectoparasite vector would in theory exhibit a household clustering. Given the absence of evidence for plague in the rat population and the observed case pattern, the bubonic plague outbreak in Glasgow is likely to be the result of human-to-human transmission, possibly by a human ectoparasite vector, as already noted by the original investigators of the outbreak.

Human ectoparasite transmission is controversial because there is very limited information about the levels of bacteraemia required for humans to transmit plague to fleas [33]. Experimental studies suggest

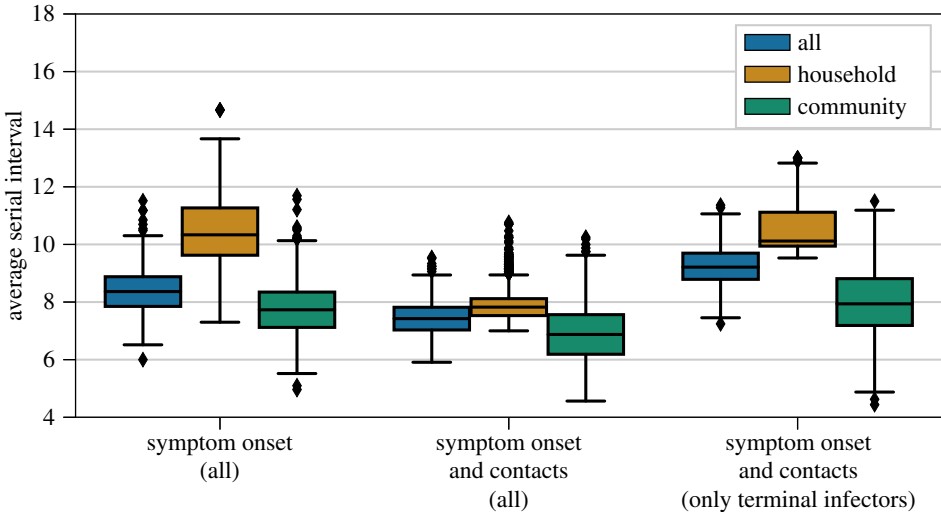

**Figure 3.** Average serial interval for all cases, community cases and household cases in 1000 simulated trees reconstructed using only the symptom onset dates, the symptom onset dates and the contact information with any infector, and the symptom onset dates and contact information with only terminally ill infectors.

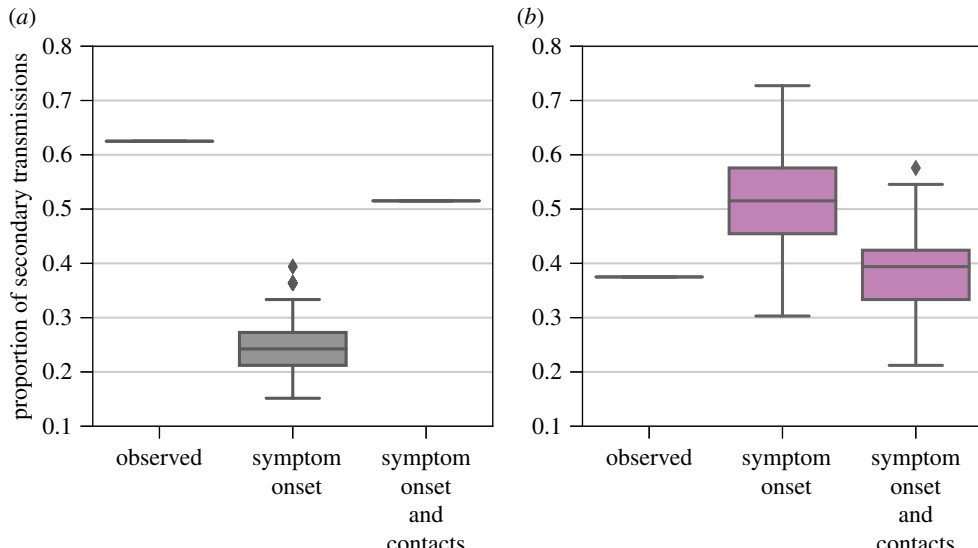

**Figure 4.** Proportion of secondary cases arising from (*a*) primary cases within the same household for observed pairs, simulated trees using only symptom onset information and simulated trees using symptom onset and contact information, and (*b*) primary cases that ultimately recovered from their infection for observed pairs, simulated trees using only symptom onset information and simulated trees using symptom onset and contact information.

that high levels of bacteraemia, consistent with terminal septicaemia, are necessary for hosts to reliably infect certain flea vectors [34]. However, we observed from the eight known pairs that three secondary transmissions occurred from two individuals who ultimately recovered; this agrees with observations that mild bacteraemia may be exhibited by individuals that are resistant to the disease or those that eventually recover [19,34,35]. Based on the above, we allowed recovered individuals to be potential infectors in one of the models. Even with this assumption, we found that the majority of secondary infections in the reconstructed trees occurred from moribund individuals, as expected. Nonetheless, individuals that survive their infections may also transmit the disease.

The likelihood-based method we used makes three assumptions about the outbreak to fully resolve the transmission trees [12]. The first assumption is that all cases during the outbreak are observed. During this outbreak, underreporting of cases is unlikely given both the thorough nature of the outbreak investigation and the overt and unequivocal course of the disease in humans. At the time of the outbreak, the symptoms for bubonic plague in humans were known, easily recognizable and cases could be confirmed with early bacteriological methods. Moreover, the plague was an extremely rare disease in Scotland at the beginning

of the twentieth century, yet officials were acutely aware of the plague pandemic spreading in India [11]. The second assumption is that all cases, excluding the index cases, are infected by another case. Humans were the only known source of the infection during the outbreak; there were no known local reservoirs for plague in Scotland and there was no evidence of plague in the rat population at the time [11]. The third condition, that the distribution of the serial interval remains stable over the course of an outbreak, is more difficult to evaluate. To our knowledge, there are no studies reporting on the temporal heterogeneity of the serial interval distribution for the plague. Thus, we consider our approach valid for the given outbreak. As shown in the sensitivity analysis, our estimates of the serial interval distribution are unchanged when the contact information is reduced, and this method is thus robust enough to deal with potential contact misclassifications.

In conclusion, our study describes an outbreak of bubonic plague in Glasgow in 1900 and uses transmission tree reconstruction to better understand the epidemiological characteristics of the outbreak. Based on the clustering of cases, bubonic plague most likely spread from human to human, possibly through a human ectoparasite vector. Without diminishing the role of rats in plague transmission during the Third Pandemic, it is important to consider that other models of transmission may apply in different historical contexts. In a modern context, the information in this study can be used to model plague outbreaks where the asymptomatic and symptomatic periods for untreated bubonic cases may be relevant.

Data accessibility. The epidemiological data used in this study are available in the report by A.K. Chalmers, 'Report on certain cases of plague occurring in Glasgow in 1900' (https://archive.org/details/b21359167/) [11]. The code used to analyse the data can be found in the supplement of Hens et al. [12].
Authors' contributions. K.R.D. and F.K. conceived and designed the study. K.R.D. performed the analysis; K.R.D., F.K. and B.V.S. interpreted the results; K.R.D. wrote the paper with input from F.K. and B.V.S. All authors gave final approval for publication.
Competing interests. The authors declare no competing interests.
Funding. K.R.D. and B.V.S. received funding by the European Research Council under the FP7-IDEAS-ERC Program (grant no. 324249). K.R.D., F.K. and B.V.S received funding from the Centre for Ecological and Evolutionary Synthesis.
Acknowledgements. We thank three anonymous reviewers and Dr Daniel R. Curtis, who provided valuable comments that improved the manuscript.

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
