## [Reviewer comments · Royal Society Open Science]

Review History

RSOS-181695.R0 (Original submission)

Review form: Reviewer 1 (Daniel Curtis)

Is the manuscript scientifically sound in its present form?

Yes

Are the interpretations and conclusions justified by the results?

Yes

Is the language acceptable?

Yes

Is it clear how to access all supporting data?

Yes

Do you have any ethical concerns with this paper?

No

Have you any concerns about statistical analyses in this paper?

No

Recommendation?

Accept with minor revision (please list in comments)

Comments to the Author(s)

This is an important paper that should be published. It is significant because (a) we still know very little about the mechanisms of historical plague transmission, and (b) it is rare we have such detailed documentary information on the dynamics of the spread and transmission process - i.e. something tangible to support abstract modelling.

Some minor adaptations:

1. Bottom p.3/top p. 4. Make clear that better understanding of epidemiological characteristics in the 3rd pandemic does not mean that these characteristics can be simply assumed to be applicable to the 2nd Pandemic. That would have to be demonstrated.
2. Next paragraph down - no evidence for rat epizootics. This is surprising to me. I want to know how unusual this is? How frequently were rat epizootics found in other late 19th- and early 20th century plagues? Basically - is Glasgow a 'special case' or representative of many other Third Pandemic cases.
3. Top of p. 12 - rather than 'historical plague outbreaks' I would explicitly note that these studies find clustering in 2nd pandemic outbreaks from the 14th to the 18th century. on that note - is there really no evidence of household clustering for 3rd pandemics elsewhere? that surprises me.
4. I would like a broader conclusion that notes that you are not dismissing the importance of the rat as a general feature of plague transmission during the 3rd pandemic, but simply suggesting here that the rat-flea-human model is not applicable for every outbreak, and other transmission models (for the bubonic version) can apply in different historical contexts.

Other outstanding problems were already suggested by the previous referees, and well addressed by the authors.

Name free to go forward: Dr. Daniel R. Curtis, Leiden University

Review form: Reviewer 2 (Joris Roosen)**Is the manuscript scientifically sound in its present form?**

Yes

Are the interpretations and conclusions justified by the results?

Yes

Is the language acceptable?

Yes

Is it clear how to access all supporting data?

Yes

Do you have any ethical concerns with this paper?

No

Have you any concerns about statistical analyses in this paper?

I do not feel qualified to assess the statistics

Recommendation?

Accept with minor revision (please list in comments)

Comments to the Author(s)

This paper provides a very interesting analysis of a small scale outbreak of plague during the 3rd pandemic in the city of Glasgow. The use of historical data to explore the likelihood of human-to-human plague transmission (through a human ectoparasite vector) and the household clustering of plague cases, make this study sufficiently original and novel for publications in "Open Science". Especially since the epidemiology of plague (both 2nd and 3rd pandemic) in Europe remain poorly understood as the authors rightfully indicate.

Since two previous referees have already commented on the analysis of the data and the conclusions derived from this analysis. And since I deem the responses of the authors on the comments made by the previous referees to be sufficiently satisfying, I will instead focus on the contextualization of the paper within the broader historiography of historical plague studies.

Afterwards, I will also include some minor comments that the authors can choose to implement or ignore at their own discretion, as I do not think them vital enough to prevent publication of the paper.

Contextualization

At several points in the paper, the authors indicate that their findings "provide important insights into the epidemiology of bubonic plague outbreaks in pre-antibiotic Europe". Although I agree that the paper provides important insights, I would ask that the authors to reflect more critically on the representativeness of findings for the 3rd pandemic as indicative for all bubonic plague outbreaks in "pre-antibiotic Europe". Just as we cannot assume that pre-industrial epidemiological experiences necessarily mirror modern ones, findings for 19th century plague outbreaks might not be so easily transposed to late medieval and early modern plague outbreaks. This may, at first, seem like a trivial point, but the disparities between the 2nd- and 3rd plague epidemics have lead some historical plague experts, such as Samuel Cohn, to claim that , "the Black Death in Europe, 1347-52, and its successive waves to the eighteenth century was any disease other than the rat-based bubonic plague, whose bacillus was discovered in 1894". Even though Cohn wrote these words before laboratory testing could conclusively prove that plague was the causative agent of the 2nd pandemic, the underlying factors that led to his provocative statement remain. Plagues of the 2nd and 3rd pandemic differed noticeably in several key epidemiological characteristics, on the issue of severity alone the present paper illustrates that very fact.

Minor comments (non-compulsory)

To gain further insight in the historical data and provide more information, I would ask the authors to take into account the following elements.

1. What was the geographical occurrence of plague within the city of Glasgow? Were cases clustered in one specific section of the city? Or were they spread out over various parts?
2. What was the total population of Glasgow in 1900. This will help understand how small a percentage of inhabitants died from plague during this outbreak (again see differences between 2nd and 3rd pandemics).

3. Page 3. For literature reference 1, an additional publication might be the book by Myron Echenberg "Plague ports: The global urban impact of bubonic plague, 1894-1901". This study also refers to the 1900 and 1901 plague outbreaks in Glasgow. Alternatively there is also the article by Echenberg: Pestis Redux, in the journal of World History, which provides some historical background information on the 1900 Glasgow outbreak.

4. Page 3. The sentence "In general, the epidemiology of plague outbreaks in Europe is poorly understood", can be linked (for the 2nd pandemic) to a recent article by Guido Alfani and Tommy Murphy (2017) "Plague and lethal epidemics in the pre-industrial world" p. 318.

5. Page 5. The authors mention several counter-measures implemented by the authorities to mitigate the impact of plague in Glasgow. Some of these 1, 2 and 4 were also widely used in the late medieval and early modern period. Given the limited number of plague cases, would the authors be willing to formulate a hypothesis regarding the effectiveness of the remaining counter-measures as decisive for the limited spread of plague?

6. Page 9. The authors mention that 60% of patients were female and that the overall case-fatality rate was 42.8%. What percentage of overall deaths were female? This also links up with page 11. In describing the epidemiological characteristics of the Glasgow outbreak, the authors reflect on several factors (case fatality rate, symptomatic period), despite the limited sample-size. Would the authors also be willing to describe female over-mortality (if this is the case) and link it up to contemporary findings of female mortality being skewed towards women in certain African cases (e.g. Tanzania)?

Decision letter (RSOS-181695.R0)

19-Nov-2018

Dear Ms Dean

On behalf of the Editors, I am pleased to inform you that your Manuscript RSOS-181695 entitled "Epidemiology of a bubonic plague outbreak in Glasgow, Scotland in 1900" has been accepted for publication in Royal Society Open Science subject to minor revision in accordance with the referee suggestions. Please find the referees' comments at the end of this email.

The reviewers and handling editors have recommended publication, but also suggest some minor revisions to your manuscript. Therefore, I invite you to respond to the comments and revise your manuscript.

- Ethics statement

- Data accessibility

It is a condition of publication that all supporting data are made available either as supplementary information or preferably in a suitable permanent repository. The data accessibility section should state where the article's supporting data can be accessed. This section

should also include details, where possible of where to access other relevant research materials such as statistical tools, protocols, software etc can be accessed. If the data has been deposited in an external repository this section should list the database, accession number and link to the DOI for all data from the article that has been made publicly available. Data sets that have been deposited in an external repository and have a DOI should also be appropriately cited in the manuscript and included in the reference list.

If you wish to submit your supporting data or code to Dryad (<http://datadryad.org/>), or modify your current submission to dryad, please use the following link:
<http://datadryad.org/submit?journalID=RSOS&manu=RSOS-181695>

- **Competing interests**

- **Authors' contributions**

- **Acknowledgements**

- **Funding statement**

Because the schedule for publication is very tight, it is a condition of publication that you submit the revised version of your manuscript before 28-Nov-2018. Please note that the revision deadline will expire at 00.00am on this date. If you do not think you will be able to meet this date please let me know immediately.

To revise your manuscript, log into <https://mc.manuscriptcentral.com/rsos> and enter your Author Centre, where you will find your manuscript title listed under "Manuscripts with Decisions". Under "Actions," click on "Create a Revision." You will be unable to make your

revisions on the originally submitted version of the manuscript. Instead, revise your manuscript and upload a new version through your Author Centre.

Once again, thank you for submitting your manuscript to Royal Society Open Science and I look

forward to receiving your revision. If you have any questions at all, please do not hesitate to get in touch.

on behalf of Dr John Dalton (Associate Editor) and Professor Kevin Padian (Subject Editor)
openscience@royalsociety.org

Reviewer comments to Author:

Reviewer: 1

Comments to the Author(s)

This is an important paper that should be published. It is significant because (a) we still know very little about the mechanisms of historical plague transmission, and (b) it is rare we have such detailed documentary information on the dynamics of the spread and transmission process - i.e. something tangible to support abstract modelling.

Some minor adaptations:

1. Bottom p.3/top p. 4. Make clear that better understanding of epidemiological characteristics in the 3rd pandemic does not mean that these characteristics can be simply assumed to be applicable to the 2nd Pandemic. That would have to be demonstrated.
2. Next paragraph down - no evidence for rat epizootics. This is surprising to me. I want to know how unusual this is? How frequently were rat epizootics found in other late 19th- and early 20th century plagues? Basically - is Glasgow a 'special case' or representative of many other Third Pandemic cases.
3. Top of p. 12 - rather than 'historical plague outbreaks' I would explicitly note that these studies find clustering in 2nd pandemic outbreaks from the 14th to the 18th century. on that note - is there really no evidence of household clustering for 3rd pandemics elsewhere? that surprises me.
4. I would like a broader conclusion that notes that you are not dismissing the importance of the rat as a general feature of plague transmission during the 3rd pandemic, but simply suggesting here that the rat-flea-human model is not applicable for every outbreak, and other transmission models (for the bubonic version) can apply in different historical contexts.

Other outstanding problems were already suggested by the previous referees, and well addressed by the authors.

Name free to go forward: Dr. Daniel R. Curtis, Leiden University

Reviewer: 2

Comments to the Author(s)

This paper provides a very interesting analysis of a small scale outbreak of plague during the 3rd pandemic in the city of Glasgow. The use of historical data to explore the likelihood of human-to-human plague transmission (through a human ectoparasite vector) and the household clustering of plague cases, make this study sufficiently original and novel for publications in "Open Science". Especially since the epidemiology of plague (both 2nd and 3rd pandemic) in Europe remain poorly understood as the authors rightfully indicate.

Since two previous referees have already commented on the analysis of the data and the

conclusions derived from this analysis. And since I deem the responses of the authors on the comments made by the previous referees to be sufficiently satisfying, I will instead focus on the contextualization of the paper within the broader historiography of historical plague studies.

Afterwards, I will also include some minor comments that the authors can choose to implement or ignore at their own discretion, as I do not think them vital enough to prevent publication of the paper.

Contextualization

At several points in the paper, the authors indicate that their findings "provide important insights into the epidemiology of bubonic plague outbreaks in pre-antibiotic Europe". Although I agree that the paper provides important insights, I would ask that the authors to reflect more critically on the representativeness of findings for the 3rd pandemic as indicative for all bubonic plague outbreaks in "pre-antibiotic Europe". Just as we cannot assume that pre-industrial epidemiological experiences necessarily mirror modern ones, findings for 19th century plague outbreaks might not be so easily transposed to late medieval and early modern plague outbreaks. This may, at first, seem like a trivial point, but the disparities between the 2nd- and 3rd plague epidemics have led some historical plague experts, such as Samuel Cohn, to claim that , "the Black Death in Europe, 1347-52, and its successive waves to the eighteenth century was any disease other than the rat-based bubonic plague, whose bacillus was discovered in 1894". Even though Cohn wrote these words before laboratory testing could conclusively prove that plague was the causative agent of the 2nd pandemic, the underlying factors that led to his provocative statement remain. Plagues of the 2nd and 3rd pandemic differed noticeably in several key epidemiological characteristics, on the issue of severity alone the present paper illustrates that very fact.

Minor comments (non-compulsory)

To gain further insight in the historical data and provide more information, I would ask the authors to take into account the following elements.

1. What was the geographical occurrence of plague within the city of Glasgow? Were cases clustered in one specific section of the city? Or were they spread out over various parts?
2. What was the total population of Glasgow in 1900. This will help understand how small a percentage of inhabitants died from plague during this outbreak (again see differences between 2nd and 3rd pandemics).
3. Page 3. For literature reference 1, an additional publication might be the book by Myron Echenberg "Plague ports: The global urban impact of bubonic plague, 1894-1901". This study also refers to the 1900 and 1901 plague outbreaks in Glasgow. Alternatively there is also the article by Echenberg: *Pestis Redux*, in the journal of *World History*, which provides some historical background information on the 1900 Glasgow outbreak.
4. Page 3. The sentence "In general, the epidemiology of plague outbreaks in Europe is poorly understood", can be linked (for the 2nd pandemic) to a recent article by Guido Alfani and Tommy Murphy (2017) "Plague and lethal epidemics in the pre-industrial world" p. 318.
5. Page 5. The authors mention several counter-measures implemented by the authorities to mitigate the impact of plague in Glasgow. Some of these 1, 2 and 4 were also widely used in the late medieval and early modern period. Given the limited number of plague cases, would the authors be willing to formulate a hypothesis regarding the effectiveness of the remaining counter-measures as decisive for the limited spread of plague?

6. Page 9. The authors mention that 60% of patients were female and that the overall case-fatality rate was 42.8%. What percentage of overall deaths were female? This also links up with page 11. In describing the epidemiological characteristics of the Glasgow outbreak, the authors reflect on several factors (case fatality rate, symptomatic period), despite the limited sample-size. Would the authors also be willing to describe female over-mortality (if this is the case) and link it up to contemporary findings of female mortality being skewed towards women in certain African cases (e.g. Tanzania)?

Author's Response to Decision Letter for (RSOS-181695.R0)

See Appendix A.

Decision letter (RSOS-181695.R1)

26-Nov-2018

Dear Ms Dean,

I am pleased to inform you that your manuscript entitled "Epidemiology of a bubonic plague outbreak in Glasgow, Scotland in 1900" is now accepted for publication in Royal Society Open Science.

on behalf of Dr John Dalton (Associate Editor) and Kevin Padian (Subject Editor)
openscience@royalsociety.org

Appendix A

Dear Dr. Dalton and Prof. Padian,

We are very pleased that our manuscript, "Epidemiology of a bubonic plague outbreak in Glasgow, Scotland in 1900," has been accepted with minor revisions in Royal Society Open Science. We thank the reviewers for their valuable comments, which have improved the manuscript. Please find below detailed responses (highlighted in bold) to the reviewers' comments.

Sincerely,

Katharine R. Dean

- Ethics- **see page 14 (Section-Ethics statement)**
- Data accessibility- **see page 14 (Section-Data accessibility)**
- Competing interests- **see page 14 (Section-Competing interests)**
- Author contributions- **see page 14 (Section- Author contributions)**
- Funding statement- **see page 14-15 (Section- Funding)**
- Acknowledgement- **see page 15 (Section- Acknowledgement)**

Reviewer comments to Author:

Reviewer: 1

Comments to the Author(s)

This is an important paper that should be published. It is significant because (a) we still know very little about the mechanisms of historical plague transmission, and (b) it is rare we have such detailed documentary information on the dynamics of the spread and transmission process - i.e. something tangible to support abstract modelling.

Some minor adaptations:

1. Bottom p.3/top p. 4. Make clear that better understanding of epidemiological characteristics in the 3rd pandemic does not mean that these characteristics can be simply assumed to be applicable to the 2nd Pandemic. That would have to be demonstrated.

We have changed this sentence to:

“Therefore, there is an opportunity to better understand the epidemiology of plague outbreaks in Europe during the Third Pandemic. Although these outbreaks cannot simply be assumed to be representative of the Second Pandemic, they can provide a valuable point of comparison for future studies.”

2. Next paragraph down - no evidence for rat epizootics. This is surprising to me. I want to know how unusual this is? How frequently were rat epizootics found in other late 19th- and early 20th century plagues? Basically - is Glasgow a 'special case' or representative of many other Third Pandemic cases.

As far as we know, the frequency of rat epizootics during this time period for other outbreaks in Europe or other parts of the world has not been the focus of a study yet, which would be a large study on its own. So, it is hard to answer how unusual the Glasgow situation was. However, we have included the numbers of plague-infected rats caught in Glasgow. This data indicates that plague-infected rats were found in later years, although not in large numbers.

3. Top of p. 12 - rather than 'historical plague outbreaks' I would explicitly note that these studies find clustering in 2nd pandemic outbreaks from the 14th to the 18th century. on that note - is there really no evidence of household clustering for 3rd pandemics elsewhere? that surprises me.

We have changed this sentence to:

“Many studies have reported household clustering of cases during Second Pandemic plague outbreaks in Europe [20-25].”

With regards to the Third Pandemic, we are not aware of any other studies that have reported an increased rate of household transmission of plague in Europe for this time period. We believe that there are very few outbreaks in Europe during the Third Pandemic that are large enough to be used in a quantitative analysis for this purpose. Moreover, without case information on the type of plague it can be difficult to know whether the cluster is bubonic cases or pneumonic cases or both. We do note that an outbreak in Nepal reports clustering of bubonic cases, while outbreaks in Bombay, Sydney, and New Orleans do not.

4. I would like a broader conclusion that notes that you are not dismissing the importance of the rat as a general feature of plague transmission during the 3rd pandemic, but simply suggesting here that the rat-flea-human model is not applicable for every outbreak, and other transmission models (for the bubonic version) can apply in different historical contexts.

We have changed the concluding paragraph to:

“In conclusion, our study describes an outbreak of bubonic plague in Glasgow in 1900 and uses transmission tree reconstruction to better understand the epidemiological characteristics of the outbreak. Based on the clustering of cases, bubonic plague most likely spread from human-to-human, possibly through a human ectoparasite vector. Without diminishing the role of rats in plague transmission during the Third Pandemic, it is important to consider that other models of transmission may apply in different historical contexts. In a modern context, the information in this study can be used to model plague outbreaks where the asymptomatic and symptomatic periods for untreated bubonic cases may be relevant.”

Reviewer: 2

Comments to the Author(s)

This paper provides a very interesting analysis of a small scale outbreak of plague during the 3rd pandemic in the city of Glasgow. The use of historical data to explore the likelihood of human-to-human plague transmission (through a human ectoparasite vector) and the household clustering of plague cases, make this study sufficiently original and novel for publications in "Open Science". Especially since the epidemiology of plague (both 2nd and 3rd pandemic) in Europe remain poorly understood as the authors rightfully indicate.

Since two previous referees have already commented on the analysis of the data and the conclusions derived from this analysis. And since I deem the responses of the authors on the comments made by the previous referees to be sufficiently satisfying, I will instead focus on the contextualization of the paper within the broader historiography of historical plague studies.

Afterwards, I will also include some minor comments that the authors can choose to implement or ignore at their own discretion, as I do not think them vital enough to prevent publication of the paper.

Contextualization

At several points in the paper, the authors indicate that their findings "provide important insights into the epidemiology of bubonic plague outbreaks in pre-antibiotic Europe". Although I agree that the paper provides important insights, I would ask that the authors to reflect more critically on the representativeness of findings for the 3rd pandemic as indicative for all bubonic plague outbreaks in "pre-antibiotic Europe".

Just as we cannot assume that pre-industrial epidemiological experiences necessarily mirror modern ones, findings for 19th century plague outbreaks might not be so easily transposed to late medieval and early modern plague outbreaks. This may, at first, seem like a trivial point, but the disparities between the 2nd- and 3rd plague epidemics have lead some

historical plague experts, such as Samuel Cohn, to claim that, "the Black Death in Europe, 1347-52, and its successive waves to the eighteenth century was any disease other than the rat-based bubonic plague, whose bacillus was discovered in 1894". Even though Cohn wrote these words before laboratory testing could conclusively prove that plague was the causative agent of the 2nd pandemic, the underlying factors that led to his provocative statement remain. Plagues of the 2nd and 3rd pandemic differed noticeably in several key epidemiological characteristics, on the issue of severity alone the present paper illustrates that very fact.

We understand the point raised here, and we have been more specific in our wording throughout the paper to make it clearer that the outbreak only represents itself and is not representative of other outbreaks in Europe. We instead say now that it provides a point of comparison for future studies.

Minor comments (non-compulsory)

To gain further insight in the historical data and provide more information, I would ask the authors to take into account the following elements.

1. What was the geographical occurrence of plague within the city of Glasgow? Were cases clustered in one specific section of the city? Or were they spread out over various parts?

We have added the sentence:

"By March 1901, the city had a population of 761,712, but the cases were primarily located in the densely-populated Gorbals area, on the south bank of the river Clyde [10]."

2. What was the total population of Glasgow in 1900. This will help understand how small a percentage of inhabitants died from plague during this outbreak (again see differences between 2nd and 3rd pandemics).

See response to point 1.

3. Page 3. For literature reference 1, an additional publication might be the book by Myron Echenberg "Plague ports: The global urban impact of bubonic plague, 1894-1901". This study also refers to the 1900 and 1901 plague outbreaks in Glasgow. Alternatively, there is also the article by Echenberg: Pestis Redux, in the journal of World History, which provides some historical background information on the 1900 Glasgow outbreak.

4. Page 3. The sentence "In general, the epidemiology of plague outbreaks in Europe is poorly understood", can be linked (for the 2nd pandemic) to a recent article by Guido Alfani and Tommy Murphy (2017) "Plague and lethal epidemics in the pre-industrial world" p. 318.

We have added the citation for Alfani and Murphy.

5. Page 5. The authors mention several counter-measures implemented by the authorities to mitigate the impact of plague in Glasgow. Some of these 1, 2 and 4 were also widely used in the late medieval and early modern period. Given the limited number of plague cases, would the authors be willing to formulate a hypothesis regarding the effectiveness of the remaining counter-measures as decisive for the limited spread of plague?

It is difficult to say which countermeasures were effective in stopping the outbreak, especially given that several were enacted and at least some (in theory) would have reduced contact events, and thus transmission. Although we think the reviewer raises an interesting point here, we hesitate to comment on it further because our aim is not to compare aspects of the Second and Third Pandemics in this paper, only to describe an outbreak and the disease in untreated bubonic plague cases.

6. Page 9. The authors mention that 60% of patients were female and that the overall case-fatality rate was 42.8%. What percentage of overall deaths were female? This also links up with page 11. In describing the epidemiological characteristics of the Glasgow outbreak, the authors reflect on several factors (case fatality rate, symptomatic period), despite the limited sample-size. Would the authors also be willing to describe female over-mortality (if this is the case) and link it up to contemporary findings of female mortality being skewed towards women in certain African cases (e.g. Tanzania)?

We have looked into this and changed the sentence in the paper to be more specific. Our data shows that 9/21 (42.8%) infected females died and 6/14 (42.8%) males died. Rather unsurprisingly, a 'test of proportions' shows that there is no difference in these rates ($p=1$) and thus no support for female over-mortality for this outbreak.